# Lactate Dehydrogenase-Elevating Virus Infection Inhibits MOG Peptide Presentation by CD11b+CD11c+ Dendritic Cells in a Mouse Model of Multiple Sclerosis

**DOI:** 10.3390/ijms25094950

**Published:** 2024-05-01

**Authors:** Pyone Pyone Soe, Mélanie Gaignage, Mohamed F. Mandour, Etienne Marbaix, Jacques Van Snick, Jean-Paul Coutelier

**Affiliations:** 1de Duve Institute, Universite Catholique de Louvain, 1200 Brussels, Belgium; pyone.soe@uclouvain.be (P.P.S.); gaignage_melanie@hotmail.com (M.G.); m.fouad4112007@gmail.com (M.F.M.); etienne.marbaix@saintluc.uclouvain.be (E.M.); 2Department of Clinical Pathology, Faculty of Medicine, Suez Canal University, Ismailia 8366004, Egypt; 3Cliniques Universitaires Saint-Luc, Université catholique de Louvain, 1200 Brussels, Belgium; 4Ludwig Institute for Cancer Research, Université Catholique de Louvain, 1200 Brussels, Belgium; jacques.vansnick@uclouvain.be

**Keywords:** experimental autoimmune encephalomyelitis, lactate dehydrogenase-elevating virus, antigen presentation, myelin oligodendrocyte glycoprotein peptide, dendritic cells, demyelination

## Abstract

Infections may affect the course of autoimmune inflammatory diseases of the central nervous system (CNS), such as multiple sclerosis (MS). Infections with lactate dehydrogenase-elevating virus (LDV) protected mice from developing experimental autoimmune encephalomyelitis (EAE), a mouse counterpart of MS. Uninfected C57BL/6 mice immunized with the myelin oligodendrocyte glycoprotein peptide (MOG35–55) experienced paralysis and lost weight at a greater rate than mice who had previously been infected with LDV. LDV infection decreased the presentation of the MOG peptide by CD11b+CD11c+ dendritic cells (DC) to pathogenic T lymphocytes. When comparing non-infected mice to infected mice, the histopathological examination of the CNS showed more areas of demyelination and CD45+ and CD3+, but not Iba1+ cell infiltration. These results suggest that the protective effect of LDV infection against EAE development is mediated by a suppression of myelin antigen presentation by a specific DC subset to autoreactive T lymphocytes. Such a mechanism might contribute to the general suppressive effect of infections on autoimmune diseases known as the hygiene hypothesis.

## 1. Introduction

Infectious agents are known to play major roles in the pathogenesis of autoimmune diseases. On the one hand, many of these autoimmune diseases, such as type 1 diabetes or MS, may be induced or at least exacerbated by infections [1,2]. On the other hand, the hygiene hypothesis suggests that common infections may decrease the probability to develop such diseases [3]. Some mechanisms responsible for these effects of infections on the development of autoimmune diseases have been analyzed in experimental models. In mice, although viruses can induce diabetes through direct pancreatic islet β cell lytic infection [4], they can also trigger or enhance anti-β cell autoimmunity through antigenic mimicry [5] or through bystander enhancement of antigen presentation [6]. Theiler’s virus triggers autoimmune CNS demyelinating disease through antigenic mimicry and epitope spreading [7,8]. Mouse hepatitis virus also induces an immune-mediated CNS demyelinating disease through macrophage and T lymphocyte activation [9].

LDV is a mouse arterivirus that induces a lifelong viremia that usually does not induce pathology, but a transient sharp pro-inflammatory response. This absence of virus-induced pathology, together with a strong modulation of the host immune microenvironment, provides a unique model to understand how a virus can modulate concomitant diseases unrelated to the infection, but with an immune component. For instance, through interferon (IFN)-γ-mediated macrophage activation, it enhances the severity of autoimmune anemia and thrombocytopenia resulting from the phagocytosis of opsonized erythrocytes and platelets [10,11]. In contrast, LDV has been reported to inhibit the development of autoimmune lupus and diabetes, through mechanisms that have so far not been elucidated [12,13]. LDV infection also prevents the development of EAE, a mouse counterpart of MS [12,13,14] that is induced in susceptible mouse strains through immunization with myelin-derived antigens, including a MOG peptide, myelin basic protein (MBP), or proteolipid protein. EAE shares several characteristics with MS, such as demyelination in white matter and neuroinflammation. The immunization of C57/Bl6 mice with peptides derived from MOG, a glycoprotein found on oligodendrocytes which participates to the structural integrity of the myelin sheath [15], triggers the activation of autoreactive CD4+ and CD8+ T lymphocytes in the CNS of immunized animals [16]. More recently, LDV has been shown to inhibit DCs through the production of type I IFN, leading to a suppression of graft-versus-host disease [17,18]. This LDV-induced type I IFN-dependent DC inhibition was later reported to lead to the suppressive effect of the virus on EAE development [19].

The purpose of this work was therefore to further analyze the effect of LDV infection, and to specify which types of DCs were affected by the infection and whether the specific T cell response against the MOG peptide was actually suppressed.

## 2. Results

### 2.1. EAE Prevention by Acute LDV Infection

To further evaluate the effect of LDV infection on EAE development, C57Bl/6 mice were infected either 1 day before or 14 days after immunization with MOG peptide. As shown in Figure 1, acute LDV infection strongly inhibited EAE development in terms of clinical symptoms (Figure 1A, *p* < 0.0001) and weight loss (Figure 1B, *p* < 0.0001). Infection 14 days after MOG peptide immunization had no effect or even enhanced clinical symptoms and weight loss (significant enhancement (*p* < 0.0001) only in one independent experiment out of two). Histologic analysis was performed 24–28 days after immunization, as inflammation in EAE typically begins in the lumbar region of the spinal cord and spreads to the entire spinal cord by the peak of disease, which usually occurs 20–27 days after immunization. Demyelination induced by MOG peptide immunization was strongly reduced (*p* = 0.0102) in mice that had been infected 1 day before, while infection 14 days after immunization had no effect (Figure 1C).

The proportion of CD45 (leukocytes, Figure 1D,E), CD3 (T lymphocytes, Figure 1F,G), and Iba1 (microglia cells, Figure 1H,I) positive cells in the spinal cord was assessed by means of immunohistochemistry in mice immunized with the MOG peptide that had been infected one day before immunization, or left uninfected. The proportion of infiltrating CD45 and CD3 positive cells was significantly reduced by acute LDV infection (*p* < 0.0001 and *p* = 0.0053, respectively). Iba1 positive cell infiltration was also slightly decreased, although not significantly (*p* = 0.1075). These results confirmed that acute LDV infection one day prior to MOG vaccination prevents the development of spinal cord inflammation associated with EAE development in contrast to delayed infection that might aggravate clinical scores.

### 2.2. Inhibition of MOG Peptide Presentation by LDV Infection

LDV has previously been shown to transiently affect conventional DC required for allogeneic presentation through a Toll-like receptor 7 (TLR7) and type 1 IFN-dependent mechanism [17,18], leading to a suppression of encephalitogenic CD4+ T cell responses [19]. To further analyze this inhibitory effect of LDV infection on antigen presentation in the EAE model, MOG-specific CD4+ T lymphocytes purified from immunized mice were cultured with antigen-presenting cells (APCs) from uninfected and LDV-infected animals and pulsed with either MOG protein or peptide. As shown in Figure 2A, T lymphocyte proliferation was decreased when APCs were derived from LDV-infected animals (*p* = 0.0159 and *p* = 0.0556, for MOG protein and MOG peptide, respectively). Similarly, IFN-γ production by these specific T lymphocytes was also moderately, although non significantly inhibited (Figure 2B; *p* = 0.0556 and *p* = 0.1508 for MOG protein and MOG peptide, respectively).

Splenocytes were subsequently separated into CD11b+CD11c+ and CD11b–CD11c+, which are both conventional DC subsets, CD11b+CD11c–, which are mainly macrophages, and CD11b–CD11c–, which are T and B cells. From those cell subsets, only CD11b+CD11c+ and CD11b–CD11c+ cells were able to trigger the proliferation of MOG-specific CD4+ T lymphocytes and IFN-γ production in the presence of antigen (Figure 2C,D). However, LDV infection suppressed only the ability of CD11b+CD11c+ cells to present antigen to specific T cells, resulting in a significant decreased proliferation (Figure 2C; *p* = 0.0003) and a slight, although not significant two-fold decrease in IFN-γ production (Figure 2D; *p* = 0.4347). 

## 3. Discussion

The increase in autoimmune and allergic diseases in industrialized countries has been linked with a decreased exposure to common infections through what has been called the hygiene hypothesis. Several mechanisms have been proposed to explain this protective effect of infections. It was first suggested that Th1-inducing pathogens that include viruses, intracellular bacteria, and parasites could suppress Th2 responses, and especially allergic diseases [20]. However, such a bias in T helper cell differentiation could not explain a protective effect against the development of autoimmune diseases, depending usually on Th1 responses, nor the protective effect of extracellular parasites that induce Th2 responses. The induction of regulatory immune responses in the course of infections, including with parasites [21], was then proposed as an additional mechanism to explain the preventive effect of infections against both allergic and autoimmune diseases [22]. 

Classical DCs have been shown to play a critical role in EAE development [23]. Among other mechanisms responsible for a modulation of concomitant immune responses, some viruses such as LDV have been shown to suppress both DC numbers and responses, leading to the prevention of T lymphocyte-dependent immune diseases including graft-versus-host disease and EAE [17,19]. That this effect of LDV on EAE resulted from a specific suppression of pathogenic anti-MOG antigen T lymphocyte activation was further demonstrated in the present study, and was shown to be linked to a specific inhibition of DC ability to present MOG peptide to these MOG-specific T lymphocytes. Together with CD11b–CD11c+CD8α+ cells, CD11b+CD11c+ cells are one of the major classical DC populations in mice, representing about 80% of spleen DCs [17]. The major function of both DC subsets is to present antigen to T lymphocytes. The role of CD11b+CD11c+ DCs in the induction of Th1 responses as well as their inhibition by viruses, but also by parasites such as *Leishmania donovani*, has been previously reported [17,24]. Our results indicate that both CD11b+CD11c+ and CD11b–CD11c+ DC subsets from uninfected animals efficiently present MOG antigen to autoreactive T lymphocytes, but that LDV infection suppresses only the ability of the CD11b+CD11c+ subset, and not of the CD11b–CD11c+ subpopulation, to efficiently activate these autoreactive T cells. The suppressive effect of viral infection on CD11b+CD11c+ cells is largely mediated by type I IFN that can be triggered by the ligation of innate receptors, including TLR7 [18,19,25]. Therefore, our results suggest that type I IFN-induced APC inhibition might be an additional mechanism that could participate to the hygiene hypothesis, as also suggested by the ability of viruses such as LDV to impair antigen presentation, but not antigen uptake, by these cells [26]. Indeed, since type I IFN is induced by most viruses as well as by some bacteria and parasites, CD11b+CD11c+ DC inhibition might be frequent enough to explain the lower prevalence of autoimmune and allergic diseases in countries where those infections are common. 

Interestingly, the same mechanism that results in the suppression of specific T lymphocyte response through the impairment of antigen presentation also triggers a general lymphocyte activation, shown by the upregulation of CD69 by these cells [25]. On the other hand, CD11b+CD11c+ DCs have also been reported to induce tolerance and to suppress EAE development [27]. This tolerogenic effect of CD11b+CD11c+ DCs was observed after the intravenous injection of MOG antigen and resulted in the secretion of IL-10 and TGFβ regulatory cytokines [27]. It is therefore quite possible that the stimulating or suppressive effect of this DC subset on anti-MOG autoimmune T lymphocyte response depends on the circumstances of their own activation.

Moreover, late infection might have a stimulating effect on EAE development, although the causative mechanisms remain to be determined. It may be postulated that at two weeks after MOG immunization, pathogenic T lymphocytes no longer require antigen presentation, but benefit from the general T lymphocyte activation dependent on the virus-induced type I IFN production [26]. Whatever the case, this difference highlights the complex relationship between viral infections and the development of autoimmune cerebral pathology of the MS type.

## 4. Materials and Methods

### 4.1. Mice

Female C57BL/6 mice (7–10 weeks) were purchased from Janvier—Bio Services B.V (Horst, The Netherlands). Depending on the experiments, 3 to 9 mice per group were used. The local ethics committee approved the study (ref. 2014/UCL/MD/008). For ethical reasons, some of the mice had to be euthanized.

### 4.2. Induction and Evaluation of EAE

To induce EAE, mice were immunized with MOG35-55 peptide (MEVGWYRSPFSRVVHYRNGK) synthesized at Ludwig Institute for Cancer Research, de Duve Institute, Université Catholique de Louvain, Brussels, Belgium, through subcutaneous injection at the base of the tail. On Day 0, the peptide was emulsified in 50 μL of Complete Freund Adjuvant (CFA) containing 400 μg of mycobacterium tuberculosis per mouse (Difco Lab., Detroit, MI, USA). On days 0 and 3, mice were given two additional doses of Bordetella pertussis toxin (300 ng per mouse) intraperitoneally (i.p.) (Enzo Life Sciences, Farmingdale, YN, USA).

To evaluate EAE development, weight loss and clinical parameters were assessed in the animals. The following clinical measures were used to determine the severity of EAE: 0 = normal; 0.5 = floppy tail; 1 = tail paralysis and mild impaired righting reflex; 2 = mild hind limb weakness and impaired righting reflex; 3 = moderate to severe hind limb paresis and/or mild forelimb weakness; 4 = complete hind limb paralysis and/or moderate to severe forelimb weakness; 5 = quadriplegia or moribund.

### 4.3. Viral Infection

Approximately 2 × 10^7^ infectious doses (ID_50_) of LDV (Riley strain; from the American Type Culture Collection, Rockville, MD, USA) were injected i.p. in 500 μL saline. Mice were infected 1 day before or 14 days after MOG35-55 peptide immunization.

### 4.4. Demyelination Analysis

At 25 to 28 days after MOG35-55 peptide immunization, groups of 3 to 6 mice were profoundly anaesthetized and perfused with 0.1 M PBS followed by 4% paraformaldehyde through the heart. Spinal cords were post-fixed in 4% paraformaldehyde overnight. Three transverse sections of the spinal cord were taken at the cervical, thoracic, and lumbar regions to examine demyelination and inflammatory cells infiltration. With a Thermo Science HM 355S Automatic Microtome (Thermo Fisher Scientific, Waltham, MA, USA), six-micrometer thick transverse sections were cut and stained with hematoxylin and eosin (H&E) and luxol fast blue (LFB). A SCN400 slide scanner was used to digitalize the images (Leica Biosystems, Wetzlar, Germany). Totaling the unstained area, dividing by the total area, and multiplying the quotient by 100 provided the percentage of demyelination.

### 4.5. Immunohistochemistry

For immunohistochemical staining, sections were deparaffinized, rehydrated, and quenched for endogenous peroxidase activity with 0.3% H_2_O_2_, retrieved antigen with 0.01 M citrate buffer, and then blocked nonspecific binding sites with Tris-HCl containing 10% normal goat serum (NGS). 

The sections were incubated overnight at 4 °C with primary antibody (CD3—Rabbit monoclonal antibody (1:250) (Thermo Fisher Scientific, Waltham, MA, USA; RM-9107-S0) (CD45—Purified Rat Anti-Mouse (1:50) (BD Bioscience, Franklin Lakes, NJ, USA; 550539) and (Iba1—Polyclonal rabbit antibody (1:400) (SynapticSystems, Göttingen, Germany, 234003) at an optimal concentration prepared in Tris-HCl containing 1% NGS. For CD 45, the slides were then incubated with Rabbit Anti-Rat IgG H&L (1:1000) (Abcam, Cambridge, UK; ab6703). The sections were incubated with HRP Labelled Polymer Anti-Rabbit Ig (Dako, Glostrup, Denmark; K 4010). Color development was performed using the 3,3′-diaminobenzidine substrate. All wash steps between incubations were with deionized water and Tris-HCl. The slides were counter-stained with hematoxylin, dehydrated, and mounted with mounting media. For the negative control, the same procedures were performed without adding the primary antibody. 

### 4.6. MOG-Specific T Cell and DC Preparation

To obtain specific MOG35-55 CD4+T cells, C57BL/6 mice were immunized with MOG35-55 peptide (100 µg/mouse) in CFA at the basis of the tail. After 7 days, para-aortic and inguinal lymph nodes were collected and CD4+ T cells were purified by means of Magnetic-Activated Cell Sorting (MACS) with positive selection.

For the in vitro functional assays, DC subpopulations were first purified by means of MACS from spleens of 24 h-LDV-infected and non-infected C57BL/6 (4 to 5 mice per group). CD11 cells were then sorted by FACS (BD FacsAria III, BD Bioscience, Franklin Lakes, NJ, USA) using PE-labeled anti-CD11b and FITC-labeled anti-CD11c.

### 4.7. In Vitro Culture

C57BL/6 adherent cells (obtained by coating 1 × 10^6^ splenocytes in a 96-well flat bottom microtiterplate for 1.5 h and removing non-adherent cells by washing the microplate twice with 37 °C PBS) and 10^4^ DC subpopulations were seeded with MOG35-55 peptide (1 µg/mL) or protein (10 µg/mL) for 2 h. Presenting cells were washed (to remove the excess of MOG) with PBS and irradiated with 30 Gy from a 137 Cs source (to prevent background proliferation and/or aspecific reaction). Purified CD4 T cells (0.15 × 10^6^/well) were incubated with DC for 48 h. 

### 4.8. Proliferation Assay

Proliferation was measured after 2 days by means of incubation with 3H-thymidine at 1 μCi (0.037 MBq)/well for a further 18 h. 3H-thymidine incorporation was measured using a scintillation counter (Packard Microplate Scintillation Counter, PerkinElmer, Waltham, MA, USA).

### 4.9. Cytokine Measurements

Cytokine production was measured after 2 days in cell culture supernatants and serum. ELISA specific for murine IFN-γ (R&D Systems, Minneapolis, MN, USA) was performed, according to the manufacturer’s instructions. Biotinylated detection Abs were used, followed by avidin-HRP (Biolegend, San Diego, CA, USA). All absorbance reads were made at 450 nm, using a 96-well plate spectrophotometer (VERSAmax, Molecular Device, San Jose, CA, USA).

### 4.10. Statistical Analysis

Prism 6 software (GraphPad Prism, La Jolla, CA, USA) was used for statistical analysis, with one-way or two-way ANOVA for multiparametric tests and Bonferroni’s post hoc tests. The Mann–Whitney test and Tukey’s multiple comparison test were also used. *p* values < 0.05 were considered to be statistically significant.

## Figures and Tables

**Figure 1 ijms-25-04950-f001:**
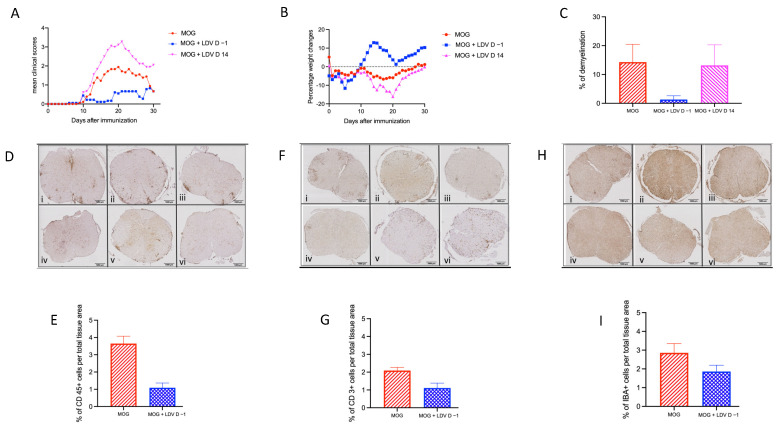
EAE prevention by LDV infection. (**A**,**B**) Groups of nine C57BL/6 mice were infected with LDV 1 day before (blue squares) or 14 days after (pink triangles) MOG peptide immunization, or left uninfected (red circles). Clinical scores (**A**) and weight (**B**) were followed daily. Results are shown for one experiment representative of two independent experiments. (**C**) Demyelination was assessed in groups of three C57BL/6 mice at 28 days after MOG peptide immunization in animals uninfected (red) or infected 1 day before (blue) or 14 days after (pink) immunization. Results are shown as means ± SEM for one experiment representative of two independent experiments. (**D**–**I**) Spinal cord sections were prepared in groups of six C57BL/6 mice at 25 days after MOG peptide immunization in non-LDV-infected mice (MOG) ((i) cervical, (ii) thoracic, and (iii) lumbar region and LDV-infected mice (LDV + MOG) ((iv) cervical, (v) thoracic, and (vi) lumbar region). The proportion of CD45 (leukocytes, (**D**,**E**)), CD3 (T lymphocytes, (**F**,**G**)), and Iba1 (microglia cells, (**H**,**I**)) positive cells in the spinal cord was assessed using a SCN400 slide scanner (Leica), at a 20× magnification (bars = 1000 µm). Results are shown as means ± SEM for one experiment representative of two independent experiments.

**Figure 2 ijms-25-04950-f002:**
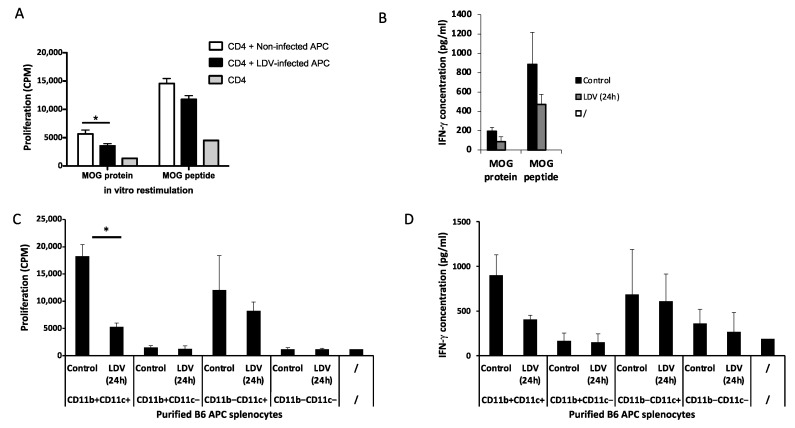
LDV inhibition of MOG peptide presentation. (**A**,**B**) Lymph node CD4+ T cells from MOG35-55-immunized C57BL/6 mice were incubated with syngeneic adherent cells previously pulsed for 2 h with MOG protein or MOG peptide. After 48 h, proliferation was measured using 3H-thymidine incorporation (**A**) and IFN-γ was measured using ELISA (**B**). Data are means ± SEM (*n* = five mice per group) of one experiment (* *p* < 0.05, Mann–Whitney test). (**C**,**D**) Lymph node CD4+ T cells from MOG-immunized C57BL/6 mice were incubated with CD11b+CD11c+, CD11b–CD11c+, CD11b+CD11c–, or CD11b–CD11c– cells purified using MACS beads and FACS sorting from LDV-infected or non-infected (control) C57BL/6 mice and pulsed for 2 h with MOG peptide. After 48 h, proliferation (**C**) and IFN-γ production (**D**) were measured. Data are means ± SEM (*n* = four mice per group) and representative of two independent experiments (* *p* < 0.05, Mann–Whitney test comparing CD11b+CD11c+ cells from LDV-infected mice and non-infected mice).

## Data Availability

The raw data supporting the conclusions of this article will be made available by the authors on request.

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
