# Peer review of "Lactate Dehydrogenase-Elevating Virus Infection Inhibits MOG Peptide Presentation by CD11b+CD11c+ Dendritic Cells in a Mouse Model of Multiple Sclerosis"

_ijms, 2024, doi:10.3390/ijms25094950_

Round 1
Reviewer 1 Report
Comments and Suggestions for Authors
Reviewer comments and suggestions
Infections with the lactate-dehydrogenase-elevating virus (LDV) prevented mice from developing Experimental Autoimmune Encephalomyelitis, a rodent version of multiple sclerosis. According to study the authors suggested that uninfected C57BL/6 mice immunized with the myelin oligodendrocyte glycoprotein peptide (MOG35–55) experienced paralysis and lost weight at a greater rate than mice who did not previously been infected with LDV. When comparing non-infected mice to infected mice, histopathological examination of the CNS showed more areas of demyelination and CD45+ and CD3+, but not Iba1+ cell infiltration. Hence the authors concluded that the protective effect of LDV infection against EAE development is mediated by a suppression of myelin antigen presentation by a specific DC subset to autoreactive T lymphocytes.
Overall, the manuscript was good. However, a few major concerns/comments needed to be explained or modified.
- General comments please check the authors guidelines for the short communication whether it is suitably arranged and for the words counts compared with the articles.
- Line 42-43 Please explain the condition briefly.
- Line 46-47 These studies needed to be well discussed, not only citation provide enough information.
- Line 47 It would be nice if the authors could explain more about EAE
- Line 52 It would be nice if the authors could discuss few points on MOG
- Figure 1 D to H should be marked with indicator for easy understanding.
- Line 114 Please mention clearly.
- Line 119-121 The first part should mention the novelty of the study rather than explaining the general sentences.
- Line 147 What does the authors want to state here, please explain it clearly.
Author Response
We thank the reviewer for her/his useful comments that were used to improve the manuscript as follows:
1. General comments please check the authors guidelines for the short communication whether it is suitably arranged and for the words counts compared with the articles.
The information on the special issue on animal research model for neurological diseases mention the possibility to submit short communications. MDPI instructions for brief reports mention a maximum of two figures and/or a table. Here we have two figures. The word count should be at least 2500 words. We have 2531 words (Introduction + Materials and Methods + Results + Discussion) + 363 word in Figure legend. The instructions ask for a text structure similar to that of a regular article, so we put the Materials and Methods section before the Results section. Abstracts of research papers should be of a maximum of 200 words, our abstract is 161 words.
2. Line 42-43 Please explain the condition briefly.
The absence of virus-induced pathology, with a strong modulation of the host immune microenvironment provides an unique model to analyse how viruses can modulate unrelated diseases. This has been developed as requested.
3. Line 46-47 These studies needed to be well discussed, not only citation provide enough information.
The reports on the effect of LDV on lupus and diabetes are old papers published 30 to 40 years ago that were descriptive and did not explain mechanisms. This is now stated.
4. Line 47 It would be nice if the authors could explain more about EAE
More information on EAE has been provided
5. Line 52 It would be nice if the authors could discuss few points on MOG
More information on MOG and on MOG peptide immunization has been added, with two additional references.
6. Figure 1 D to H should be marked with indicator for easy understanding.
Scale bars have been added in Fig 1D, F and H.
7. Line 114 Please mention clearly.
The sentence has been rephrased to better define CD11b+CD11c+ DC.
8. Line 119-121 The first part should mention the novelty of the study rather than explaining the general sentences.
A sentence to emphasize the novelty of our results has been added.
9. Line 147 What does the authors want to state here, please explain it clearly.
The part of the protocol for EAE induction and evaluation have been more clearly separated (now in the point 2.2 of the Material and Methods section).
Reviewer 2 Report
Comments and Suggestions for Authors
The manuscript is well written and the contents are correct. However, some changes are recommended in order to improve the final quality and be published.
Elemental changes:
This work would be more understandable if the material and methods section were before the results section. In this material and methods section, the number of mice, as well as the experimental time (induction time or period from the induction of the experimental disease to the time of analysis of biological samples) should be easily identifiable. As the manuscript is presented, these aspects can only be deduced from the figure legends in Figure 2 and Figure 1 (in that order).
Secondary changes:
In the introduction section, lines 33 and 34 begin with the same expression (on the other hand). In order to avoid being repetitive, it is recommended that another equivalent expression be used at the beginning of one of the sentences.
In section 2.2. in line 90 the acronym TLR7 is written. The first time an acronym appears, it would be advisable to explain what it belongs to (Toll-like receptor 7).
Author Response
We thank the reviewer for her/his useful comments that helped to improve our manuscript.
Specificaly, we responded to the comments as follows:
This work would be more understandable if the material and methods section were before the results section. In this material and methods section, the number of mice, as well as the experimental time (induction time or period from the induction of the experimental disease to the time of analysis of biological samples) should be easily identifiable. As the manuscript is presented, these aspects can only be deduced from the figure legends in Figure 2 and Figure 1 (in that order).
The Material and Method section was placed before the Results section, as required. Number of mice and times or periods were added.
In the introduction section, lines 33 and 34 begin with the same expression (on the other hand). In order to avoid being repetitive, it is recommended that another equivalent expression be used at the beginning of one of the sentences.
On line 33, the sentence starts with "On the one hand", that is contrasted with the second sentence (On the other hand). This was used to mark the contrast between these two different types of observations.
In section 2.2. in line 90 the acronym TLR7 is written. The first time an acronym appears, it would be advisable to explain what it belongs to (Toll-like receptor 7).
The full name was added.